# Towards Theoretically Inspired Neural Initialization Optimization

**Yibo Yang[1], Hong Wang[2], Haobo Yuan[3], Zhouchen Lin[2,4,5*]**

[1]JD Explore Academy, Beijing, China
[2]Key Lab. of Machine Perception (MoE), School of Intelligence Science and Technology, Peking University
[3]Institute of Artificial Intelligence and School of Computer Science, Wuhan University
[4]Institute for Artificial Intelligence, Peking University
[5]Pazhou Laboratory, Guangzhou, China

## Abstract

Automated machine learning has been widely explored to reduce human efforts in designing neural architectures and looking for proper hyperparameters. In the domain of neural initialization, however, similar automated techniques have rarely been studied. Most existing initialization methods are handcrafted and highly dependent on specific architectures. In this paper, we propose a differentiable quantity, named GradCosine, with theoretical insights to evaluate the initial state of a neural network. Specifically, GradCosine is the cosine similarity of sample-wise gradients with respect to the initialized parameters. By analyzing the sample-wise optimization landscape, we show that both the training and test performance of a network can be improved by maximizing GradCosine under gradient norm constraint. Based on this observation, we further propose the neural initialization optimization (NIO) algorithm. Generalized from the sample-wise analysis into the real batch setting, NIO is able to automatically look for a better initialization with negligible cost compared with the training time. With NIO, we improve the classification performance of a variety of neural architectures on CIFAR-10, CIFAR-100, and ImageNet. Moreover, we find that our method can even help to train large vision Transformer architecture without warmup.

## 1 Introduction

For a deep neural network, architecture [19, 20, 21] and parameter initialization [18, 16] are two initial elements that largely account for the final model performance. Lots of human efforts have been devoted to finding better answers with respect to the two aspects. To automatically produce better architectures, neural architecture search [62, 38, 5] has been a research focus. However, on the other hand, using automated techniques for parameter initialization has rarely been studied.

Previous initialization methods are mostly handcrafted. They focus on finding proper variance patterns of randomly initialized weights [18, 16, 40, 36] or rely on empirical evidence derived from certain architectures [58, 23, 15, 7]. Recently, [60, 8] propose learning-based initialization that learns to tune the norms of the initial weights so as to minimize a quantity that is intimately related to favorable training dynamics. Despite being architecture-agnostic and gradient-based, these methods do not consider sample-wise landscape. Their derived quantities lack theoretical supports for being related to model performance. It is unclear whether optimizing these quantities can indeed lead to better training or generalization performance.

In order to find a better initialization, a theoretically sound quantity that intimately evaluates both the training and test performance should be designed. Finding such a quantity is a non-trivial problem and

---

*: corresponding author.

36th Conference on Neural Information Processing Systems (NeurIPS 2022).

shows many challenges. First, the test performance is mostly decided by the converged parameters after training, while the training dynamic is more related to the parameters at initialization or during training. Second, to efficiently find a better starting point, the quantity is expected to be differentiable to enable the optimization in the continuous parameter space. To this end, we leverage the recent advances in optimization landscape analysis [2] to propose a novel differentiable quantity and develop a corresponding algorithm for its optimization at initialization.

Specifically, our quantity is inspired by analyzing the optimization landscapes of individual training samples [59]. Through generalizing prior theoretical results on batch-wise optimization [2] to sample-wise optimization, we prove that both the network's training and generalization error are upper bounded by a theoretical quantity that correlates with the cosine similarity of the sample-wise local optima. Moreover, this quantity also relates to the training dynamic since it reflects the optimization path consistency [35, 37] from the starting point. Unfortunately, the sample-wise local optima are intractable. With the hypothesis that the sample-wise local optima can be reached by the first-order approximation from the initial parameters, we can approximate the quantity via the sample-wise gradients at initialization. Our final result shows that, under a limited gradient norm, both the training and test performance of a network can be improved by maximizing the cosine similarity of sample-wise gradients, named GradCosine, which is differentiable and easy to implement.

We then propose the Neural Initialization Optimization (NIO) algorithm based on GradCosine to find a better initialization agnostic of architecture. We generalize the algorithm from the sample-wise analysis into the batch-wise setting by dividing a batch into sub-batches for friendly implementation. We follow [8, 60] using gradient descent to learn a set of scalar coefficients of the initialized parameters. These coefficients are optimized to maximize GradCosine for better training dynamic and expected performance while constraining the gradient norm from explosion.

Experiments show that for a variety of deep architectures including ResNet [19], DenseNet [21], and WideResNet [56], our method achieves better classification results on CIFAR-10/100 [27] than prior heuristic [18] and learning-based [8, 60] initialization methods. We can also initialize ResNet-50 [19] on ImageNet [9] for better performance. Moreover, our method is able to help the recently proposed Swin-Transformer [32] achieve stable training and competitive results on ImageNet even without warmup [17], which is crucial for the successful training of Transformer architectures [31, 52].

## 2 Related Work

### 2.1 Network Initialization

Existing initialization methods are designed to control the norms of network parameters via Gaussian initialization [16, 18] or orthonormal matrix initialization [40, 36] with different variance patterns. These analyses are most effective for simple feed-forward networks without skip connections or normalization layers. Recently, initialization techniques specified for some complex architectures are proposed. For example, [58] studied how to initialize networks with skip connections and [23] generalized the results into Transformer architecture [49]. However, these heuristic methods are restricted to specific architectures. Automated machine learning has achieved success in looking for hyperparameters [3, 13] and architectures [62, 38, 5, 54, 55, 22], while similar techniques for neural network initialization deserve more exploration. Current learning based initialization methods [8, 60] optimize the curvature [8] or the loss reduction of the first stochastic step [60] using gradient descent to tune the norms of the initial parameters. However, these methods lack theoretical foundations to be related to the model performance. Different from these methods, our proposed GradCosine is derived from a theoretical quantity that is the upper bound of both training and generalization error.

### 2.2 Evaluating Model Performance at Initialization

Evaluating the performance of a network at initialization has been an important challenge and widely applied in zero-shot neural architecture search [1, 34, 6, 42, 59, 41, 29] and pruning [50, 28, 44]. The evaluation quantities in these studies are mainly based on the initial gradient norm [44, 41], the eigenvalues of neural tangent kernel [6, 41], and the Fisher information matrix [47, 48, 45]. However, these quantities cannot reflect optimization landscape, which is crucial for training dynamic and generalization [4, 12, 14, 43, 30, 2]. [2] provided theoretical evidence that for a sufficiently large neighborhood of a random initialization, the optimization landscape is nearly convex and semi-

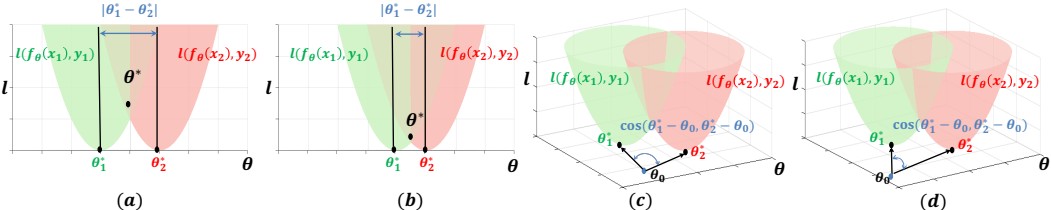

Figure 1: (a) Optimization landscape with sparser sample-wise local optima corresponds to a worse $\theta^*$ (larger loss $l$). (b) Optimization landscape with denser sample-wise local optima corresponds to a better $\theta^*$ (smaller loss $l$). However, the density of sample-wise local optima cannot reflect training path. We further leverage the cosine similarity of sample-wise local optima. Under the same local optima density, (d) corresponds to a better training dynamic than (c), since (d) enjoys a better optimization path consistency (smaller cosine distance between $\theta_1^* - \theta_0$ and $\theta_2^* - \theta_0$). We give more detailed explanations and discussions in Appendix C.

smooth. Based on the result, [59] proposed to use the density of sample-wise local optima to evaluate and rank neural architectures. Our study also performs sample-wise landscape analysis, but differs from [59] in that our proposed quantity is differentiable and reflects optimization path consistency, while the quantity in [59] is non-differentiable so cannot serve for initialization optimization. We make a comparison in more details in Appendix C.

## 3 Theoretical Foundations

### 3.1 Sample-wise Optimization Landscape Analysis

Conventional optimization landscape analyses [30, 2] mainly focus on the objective across a mini-batch of samples and miss potential evidence hidden in the optimization landscapes of individual samples. As recently pointed out in [59], by decomposing a mini-batch objective into the summation of sample-wise objectives across individual samples in the mini-batch, they find that the network with denser sample-wise local optima tends to reach a better local optima for the mini-batch objective, as shown in the Figure (1) (a)-(b). Based on this insight, they propose to use sample-wise local optima density to judge model performance at initialization.

**Density of sample-wise local optima.** For a batch of training samples $S = \{(x_i, y_i)\}_{i \in [n]}$, a loss function $l(\hat{y}_i, y_i)$, and a network $f_\theta(\cdot)$ parameterized by $\theta \in \mathbb{R}^m$, the sample-wise local optima density $\Psi_{S,l}(f_{\theta_0}(\cdot))$ is measured by the averaged Manhattan distance between the pair-wise local optima $\{\theta_i^*\}_{i \in [n]}$ of all $n$ samples near the random initialization $\theta_0$ [59], *i.e.*,

$$\Psi_{S,l}(f_{\theta_0}(\cdot)) = \frac{\sqrt{\mathcal{H}}}{n^2} \sum_{i,j} ||\theta_i^* - \theta_j^*||_1, \ \ i, j \in [1, n], \tag{1}$$

where $\mathcal{H}$ is the smoothness upper bound: $\forall k \in [m], i \in [n], [\nabla^2 l(f_\theta(x_i), y_i)]_{k,k} \leq \mathcal{H}$ and $\mathcal{H}$ always exists with the smoothness assumption that for a neighborhood $\Gamma_{\theta_0}$ of a random initialization $\theta_0$, the sample-wise optimization landscapes are nearly convex and semi-smooth [2]. Based on this assumption, the training error of the network can be upper bounded by $\frac{n^3}{2}\Psi_{S,l}^2(f_{\theta_0}(\cdot))$. Moreover, with probability $1-\delta$, the generalization error measured by population loss $\mathbb{E}_{(x_u, y_u) \sim \mathcal{D}}[l(f_\theta^*(x_u), y_u)]$ is upper bounded by $\frac{n^3}{2}\Psi_{S,l}^2(f_{\theta_0}(\cdot)) + \frac{\sigma}{\sqrt{n\delta}}$, where $\mathcal{D} = \{(x_u, y_u)\}_{u \in [U]}$ is the underlying data distribution of test samples and $\sigma^2$ is the upper bound of $Var_{(x_u, y_u) \sim \mathcal{D}}[||\theta^* - \theta_u^*||_1^2]$ [59].

Although the sample-wise local optima density is theoretically related to both the network training and generalization error, we point out that it may not be a suitable quantity to evaluate network initialization due to the following reasons. First, it ignores the optimization path consistency from initialization $\theta_0$ to each sample-wise local optimum $\theta_i^*$. As shown in Figure 1 (c)-(d), while both (c) and (d) have the same local optima density, in Figure 1 (d) the optimization path from the initialization

to the local optima of the two samples are more consistent. Training networks on samples with more consistent optimization paths using batch gradient descent naturally corresponds to a better training dynamic that enjoys faster and more stable convergence [39, 37]. Second, the sample-wise local optima in Eq. (1) are intractable. It can be approximated by measuring the consistency of sample-wise gradient signs [59]. But it is non-differentiable and cannot serve for initialization optimization.

Based on this observation, we aim to derive a new quantity that directly reflects optimization path consistency and is a differentiable function of the initialization $\theta_0$. Our proposed quantity is based on the cosine similarity of the paths from initialization to sample-wise local optima.

**Cosine similarity of sample-wise local optima.** Concretely, our quantity can be formulated as:

$$\Theta_{S,l}(f_{\theta_0}(\cdot)) = \frac{\mathcal{H}\alpha^2}{n} \sum_{i,j} \left( \frac{\alpha}{\beta} - \cos\angle(\theta_i^* - \theta_0, \theta_j^* - \theta_0) \right), \; i,j \in [1,n], \tag{2}$$

where $\alpha$ and $\beta$ are the maximal and minimal $\ell_2$-norms of the sample-wise optimization paths, *i.e.,* $\alpha = \max(||\theta_i^* - \theta_0||_2), \beta = \min(||\theta_i^* - \theta_0||_2), \forall i \in [1,n]$, and $\cos\angle(\theta_i^* - \theta_0, \theta_j^* - \theta_0)$ refers to the cosine similarity of the paths from initialization $\theta_0$ to sample-wise local optima, $\theta_i^*$ and $\theta_j^*$. The cosine term in Eq. (2) reflects the optimization path consistency. Together with the distance term $\frac{\alpha}{\beta}$, it is also able to measure the density of sample-wise local optima. In the ideal case, when all the local optima are located at the same point, $\Theta_{S,l}(f_{\theta_0}(\cdot)) = \Psi_{S,l}(f_{\theta_0}(\cdot)) = 0$. Hence, compared with $\Psi$, our $\Theta$ is more suitable for evaluating the initialization quality.

## 3.2 Main Results

In this subsection, we theoretically illustrate how minimizing the quantity $\Theta$ in Eq. (2) corresponds to better training and generalization performance. Similar to [59], our derivations are also based on the evidence that there exists a neighborhood for a random initialization such that the sample-wise optimization landscapes are nearly convex and semi-smooth [2].

**Lemma 1.** *There exists no saddle point in a sample-wise optimization landscape and every local optimum is a global optimum [59].*

Based on Lemma 1, we can draw a relation between the training error and $\Theta_{S,l}(f_{\theta_0}(\cdot))$. Moreover, we show that the proposed quantity is also related to generalization performance as the upper bound of population error. We present the following two theoretical results.

**Theorem 2.** *The training loss $\mathcal{L} = \frac{1}{n} \sum_i l(f_{\theta^*}(x_i), y_i)$ of a trained network $f_{\theta^*}$ on a dataset $S = \{(x_i, y_i)\}_{i \in [n]}$ is upper bounded by $\Theta_{S,l}(f_{\theta_0}(\cdot))$, and the bound is tight when $\Theta_{S,l}(f_{\theta_0}(\cdot)) = 0$.*

**Theorem 3.** *Suppose that $\sigma^2$ is the upper bound of $Var_{(x_u,y_u)\sim\mathcal{D}}[||\theta^* - \theta_u^*||_2^2]$, where $\theta_u^*$ is the local optimum in the convex neighborhood of $\theta_0$ for test sample $(x_u, y_u)$. With probability $1 - \delta$, the population loss $\mathbb{E}_{(x_u,y_u)\sim\mathcal{D}}[l(f_{\theta^*}(x_u), y_u)]$ is upper bounded by $\Theta_{S,l}(f_{\theta_0}(\cdot)) + \frac{\sigma}{\sqrt{n\delta}}$.*

We provide proofs of these two theorems in Appendix A. Combining both theorems, we can conclude that $\Theta_{S,l}(f_{\theta_0}(\cdot))$ is the upper bound of both training and generalization errors of the network $f_{\theta^*}$. Therefore, minimizing $\Theta$ theoretically helps to improve the model performance.

Albeit theoretically sound, $\Theta_{S,l}(f_{\theta_0}(\cdot))$ requires sample-wise optima $\theta_i^*$ which are intractable at initialization. To this end, we will show how to develop a differentiable and tractable objective based on Eq. (2) in Section 4, and introduce the initialization optimization algorithm in Section 5.

## 4 GradCosine

### 4.1 First-Order Approximation of Sample-Wise Optimization

Since we are dealing with sample-wise optimization, it is reasonable to calculate each sample-wise optimum by the first-order approximation. We hypothesize that each sample-wise optimum can be reached via only one-step gradient descent from the initialized parameters. Its rationale lies in that it is very easy for a deep neural network to learn the optimum for only one training sample with gradient descent. Based on this hypothesis, we can approximate each local optimum as:

$$\theta_i^* \approx \theta_0 - \eta g_i, \; i \in [1,n], \tag{3}$$

---

**Algorithm 1** GradCosine (GC) and gradient norm (GN)

---

**Require:** Initial network parameters $\theta_0$, and a batch of samples $S = \{(x_i, y_i)\}_{i \in [B]}$, $B = |S|$.

1: **for** i=1 to $B$ **do**
2:     Compute the sample-wise gradient:
      $g_i \leftarrow \nabla_\theta l(f_\theta(x_i), y_i)|_{\theta_0}; g_i \in \mathbb{R}^m$
3: **end for**
4: Compute the average of gradient norm:
    $\mathbf{GN}(S, \theta_0) \leftarrow \frac{1}{B} \sum_{i=1}^{B} ||g_i||_2$
5: Compute the cosine similarity of gradients:
    $\phi_{i,j} \leftarrow \frac{g_i \cdot g_j}{||g_i||_2 \cdot ||g_j||_2}, i = 1, ..., B, j = 1, ..., B$
6: Compute the average of gradient cosine:
    $\mathbf{GC}(S, \theta_0) \leftarrow \frac{1}{B^2} \sum_{i=1}^{B} \sum_{j=1}^{B} \phi_{i,j}$
7: Output **GN** and **GC**.

---

where $g_i = \nabla_\theta l(f_\theta(x_i), y_i)|_{\theta_0}$ is the sample-wise gradient at initialization, and $\eta$ is the learning rate. With Eq. (3), the upper bound quantity in Eq. (2) can be simplified as:

$$\Theta \approx \frac{\mathcal{H} g_{max}^2}{n} \sum_{i,j} \left( \frac{g_{max}}{g_{min}} - \cos \angle(g_i, g_j) \right), \quad i, j \in [1, n], \tag{4}$$

where $g_{max}$ and $g_{min}$ are the maximal and minimal sample-wise gradient norms at initialization, *i.e.*, $g_{max} = \max(||g_i||_2), g_{min} = \min(||g_i||_2), \forall i \in [1, n]$.

### 4.2 Guidance for Initialization

Suppose that the sample-wise gradient at initialization is upper bounded by a constant $\gamma$, *i.e.*, $g_{max} \leq \gamma$, and then we have:

$$\Theta \leq \frac{\mathcal{H} \gamma^2}{n} \sum_{i,j} \left( \frac{\gamma}{g_{min}} - \cos \angle(g_i, g_j) \right), \quad i, j \in [1, n]. \tag{5}$$

From Eq. (2) to Eq. (5), we have almost converted the upper bound into a quantity that can be easily calculated by the sample-wise gradient at initialization, except for the term $g_{min}$ and the constraint $g_{max} \leq \gamma$. We will explicitly add corresponding optimization objective and constraint in our neural initialization optimization algorithm in Section 5 to ensure a small $\gamma/g_{min}$.

From Eq. (5), we can draw some useful guidance for a better initialization:

(1) The sample-wise gradient norms of the initialized parameters should be large and close to the maximal value bounded by $\gamma$ to induce a small $\gamma/g_{min}$;

(2) The cosine similarity of the sample-wise gradients should be as large as possible.

**Relations to Favorable Training Dynamic.** There are significant evidences in existing studies that support the two rules for better training dynamic. The first rule is intimately related to the neural tangent kernel (NTK) [25] that has recently been shown to determine the early learning behavior [51, 41, 6, 42]. Specifically, [41] finds that the training dynamic of a neural network can be characterized by the trace norm of NTK at initialization and further approximates it via the initial gradient norm as a gauge to search for neural architectures. [60] initializes a network by maximizing the loss reduction of the first gradient descent step, which also corresponds to gradient norm in the first-order approximation of the loss function. The two rules together prefer a model whose sample-wise gradients have close $\ell_2$ and cosine distances. It is in line with prior observations that the initial gradient variance should be small to enable a large learning rate [31, 61]. However, gradient variance is not a proper optimization objective due to its high sensitivity to gradient norm.

While the first rule of improving the initial gradient norm has been suggested for better training dynamic in prior studies [60, 41], the second rule is completely new. We refer this novel quantity, *i.e.,* $\cos \angle(g_i, g_j)$, for evaluating the network initialization as GradCosine. Intuitively, in the ideal case where the initial gradients of all samples are identical, we have the smallest upper bound of training and generalization error. Moreover, both gradient norm and GradCosine are differentiable and thus enable optimization to find a better initialization. We provide the calculation of gradient norm (GN) and GradCosine (GC) for an initialized network in Algorithm 1.

**Algorithm 2** Batch GradCosine (B-GC) and batch gradient norm (B-GN)

---

**Require:** Initialized network parameters $\theta_0$, a batch of samples $S = \{(x_i, y_i)\}_{i \in [B]}$, the number of sub-batches $D$, and overlap ratio $r$.
 1: **for** $d = 1$ to $D$ **do**
 2:     Compute the batch-wise gradient:
        $g_d \leftarrow \frac{1}{N} \sum_{j \in S_d} \nabla_\theta l(f_\theta(x_j), y_j)|_{\theta_0}$
 3: **end for**
 4: Compute **B-GN**$(S, \theta_0)$ and **B-GC**$(S, \theta_0)$ with batch-wise gradients following Lines (4-6) in Algorithm 1;
 5: Output **B-GN** and **B-GC**.

---

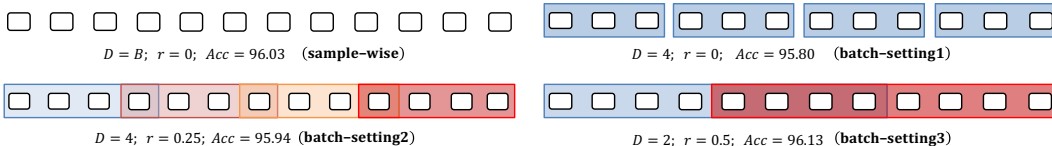

Figure 2: An illustration of sample-wise inputs and batch-wise inputs with different batch settings. The example experiment is conducted on CIFAR-10 with ResNet-110.

## 5 Neural Initialization Optimization

Based on the GradCosine, we propose our Neural Initialization Optimization (NIO) algorithm. Our algorithm follows [60, 8] to rectify the variance of the initialized parameters in a network via gradient descent. Concretely, we introduce a set of learnable scalar coefficients denoted as $M = \{\omega_1, \cdots, \omega_m\}$. The initial parameters $\theta_0 = \{W_1, \cdots, W_m\}$ rectified by these coefficients can be formulated as $\theta_M = \{\omega_1 W_1, \cdots, \omega_m W_m\}$. The NIO algorithm mainly solves the following constrained optimization problem:

$$\begin{aligned} \underset{M}{\text{maximize}} \quad & \text{GC}(S, \theta_M) + \text{GN}(S, \theta_M), \\ \text{subject to} \quad & \max_i(\|g_i\|_2) \leq \gamma, \;\; i = 1, ..., B, \end{aligned} \tag{6}$$

where GC and GN refer to GradCosine and gradient norm, respectively, as calculated in Algorithm 1, and $B$ is the batchsize of the batch data $S$. With the maximal sample-wise gradient norm bounded by $\gamma$, we maximize both GradCosine and gradient norm.

Although the problem in Eq. (6) is differentiable and tractable, inferring GradCosine and gradient norm in the sample-wise manner can be very time-consuming especially for networks trained on large datasets such as ImageNet. Moreover, using gradient-based methods to solve Eq. (6) requires deriving the second-order gradient with respect to the network parameters, which leads to unbearable memory consumption in the sample-wise manner. To this end, we introduce the batch-wise relaxation.

### 5.1 Generalizing to Batch Settings

To reduce the time and space complexity of gradient calculation, we further generalize GC and GN from the sample-wise manner into a more efficient batch setting. Specifically, for a mini-batch of training samples $S = \{(x_i, y_i)\}_{i \in [B]}$, we split it into $D$ sub-batches $S_1, ..., S_d, ..., S_D$ with an overlap ratio of $r$ that is used to relieve the violation of the sample-wise setting. The number of samples in each sub-batch is $N = \lceil \frac{B}{D-r} \rceil$, and each sub-batch can be denoted as:

$$S_d = \{(x_i, y_i)\}_{i = N(d-1)(1-r)+1, ..., N(d-1)(1-r)+N}, \tag{7}$$

where $d = 1, ..., D$. The sample-wise input can be regarded as the special case of the batch split when setting $D = B, r = 0$. We further illustrate some split examples in Figure 2. We empirically find that making the sub-batches overlapped with each other stabilize the optimization and results in better initialization. As shown in Figure 2, the test accuracies of the models trained using our initialization with different batch settings further confirm this practice.

The GradCosine and gradient norm under the batch setting are illustrated in Algorithm 2. The main difference is that, in the batch setting, the gradient is computed as the average of gradients in a

---

**Algorithm 3** Neural Initialization Optimization

---

**Require:** Initialized parameters $\theta_0 \in \mathbb{R}^m$, learning rate $\tau$ for scale coefficients $M$, upper bound of gradient norm $\gamma$, lower bound of the scale coefficients $\underline{\alpha} = 0.01$, total iterations $T$, batch size $B$, the number of sub-batches $D$, and overlap ratio $r$.

1:  $M^{(1)} \leftarrow \mathbf{1}$, where $\mathbf{1}$ denotes the all-ones vector in $\mathbb{R}^m$;
2:  **for** $t = 1$ to $T$ **do**
3:      Sample $S_t$ from the training set;
4:      Compute $g_i$, B-GC$(S_t, \theta_{M^{(t)}})$, and B-GN$(S_t, \theta_{M^{(t)}})$ by Algorithm 2;
5:      Compute the $g_{max} = \max_i(||g_i||_2), i = 1, ..., D$;
6:      **if** $g_{max} > \gamma$ **then**
7:          $M^{(t+1)} \leftarrow M^{(t)} - \tau \nabla_{M^{(t)}}$B-GN
8:      **else**
9:          $M^{(t+1)} \leftarrow M^{(t)} + \tau \nabla_{M^{(t)}}($B-GC $+$ B-GN$)$
10:      **end if**
11:      Clamp $M^{(t+1)}$ using $\underline{\alpha}$;
12: **end for**
13: Output the rectified initialization parameters $\theta_M{}^{(T)}$.

---

sub-batch. Our optimization problem in the batch setting can be formulated as:

$$
\begin{aligned}
\underset{M}{\text{maximize}} \quad & \text{B-GC}(S, \theta_M) + \text{B-GN}(S, \theta_M), \\
\text{subject to} \quad & \max_i(||g_i||_2) \le \gamma, \;\; i = 1, ..., D,
\end{aligned}
\tag{8}
$$

where B-GC and B-GN refer to the batch-wise GradCosine and gradient norm as calculated in Algorithm 2.

### 5.2  Main Algorithm

The final neural initialization optimization is illustrated in Algorithm 3. It rectifies the initialized parameters by gradient descent to solve the constrained optimization problem in Eq. (8).

Concretely, we iterate for $T$ iterations. At each iteration, a random batch data $S_t$ is sampled from the training set, and divided into sub-batches according to $D$ and $r$. We calculate B-GN and B-GC by Algorithm 2. If the maximal gradient norm is larger than the predefined upper bound $\gamma$, we minimize the averaged batch gradient norm to avoid gradient explosion at initialization. When the constraint is satisfied, we maximize the GradCosine (B-GC) and gradient norm (B-GN) objectives simultaneously, in order to minimize the upper bound of the quantity $\Theta$ in Eq. (5), which intimately corresponds to not only a lower training and generalization error, but also a better training dynamic.

## 6  Experiments

### 6.1  Initialization for CNN

**Dataset and Architectures.** We validate our method on three widely used datasets including CIFAR10/100 [27] and ImageNet [9]. We select three kinds of convolution neural networks including ResNet [19], DenseNet [21], and WideResNet [56]. On CIFAR10/100, we adopt ResNet110, DenseNet100, and the 28-layer Wide ResNet with Widen Factor 10 (WRN-28-10) as three main architectures for evaluation. To further show that our initialization method helps training dynamic for better training stability, we also conduct experiments on the same architectures but remove the batch normalization (BN) [24] layers. Moreover, to illustrate that our method can be extensible to large-scale benchmarks, we test our proposed NIO on ImageNet using ResNet-50. The detailed settings for different architectures and datasets are described in Appendix B.

**Experiment Results on CIFAR-10/100.** We select four different initialization methods for comparison: (1) Kaiming Initialization [18]; (2) First train the network for one epoch with a linear warmup learning rate, denoted as Warmup; (3) MetaInit [8]; and (4) GradInit [60]. (3) and (4) are learning-based initialization as ours. We re-implement their methods using the code provided in [60]. For MetaInit, GradInit, and our proposed NIO, the initialization is rectified based on the Kaiming initialized parameters. After initialization, we train these models for 500 epochs with the same

Table 1: Test accuracies of three architectures on CIFAR-10. Best results are marked in bold.

| Model | ResNet-110 | DenseNet-100 | WRN-28-10 |
|---|---|---|---|
| | w/ BN | | |
| Kaiming | 95.53± 0.19 | 95.75± 0.13 | 97.27± 0.27 |
| Warmup | 95.56± 0.12 | 95.73± 0.27 | 97.30± 0.18 |
| Metainit | 95.45± 0.33 | 95.75± 0.11 | 97.26± 0.17 |
| Gradinit | 95.81± 0.29 | 95.77± 0.22 | 97.34± 0.15 |
| NIO | **96.13**± 0.16 | **96.07**± 0.18 | **97.50**± 0.21 |
| | w/o BN | | |
| Kaiming | 94.83± 0.24 | 94.78± 0.16 | 97.15± 0.33 |
| Warmup | 94.75± 0.21 | 94.86± 0.15 | 97.20± 0.18 |
| Metainit | 94.79± 0.15 | 95.15± 0.19 | 97.27± 0.25 |
| Gradinit | 95.12± 0.16 | 95.31± 0.37 | 97.12± 0.13 |
| NIO | **95.27**± 0.19 | **95.62**± 0.18 | **97.35**± 0.19 |

Table 2: Test accuracies of three architectures on CIFAR-100. Best results are marked in bold.

| Model | ResNet-110 | DenseNet-100 | WRN-28-10 |
|---|---|---|---|
| | w/ BN | | |
| Kaiming | 74.54± 0.21 | 76.58± 0.23 | 81.40± 0.28 |
| Warmup | 74.63± 0.33 | 76.60± 0.32 | 81.45± 0.25 |
| Metainit | 74.32± 0.26 | 76.23± 0.28 | 81.46± 0.20 |
| Gradinit | 75.40± 0.17 | 76.14± 0.21 | 81.35± 0.31 |
| NIO | **75.72**± 0.15 | **76.86**± 0.26 | **81.83**± 0.20 |
| | w/o BN | | |
| Kaiming | 73.03± 0.23 | 71.27± 0.25 | 79.28± 0.26 |
| Warmup | 73.10± 0.14 | 71.50± 0.24 | 79.58± 0.13 |
| Metainit | 72.60± 0.17 | 71.68± 0.37 | 79.15± 0.24 |
| Gradinit | 72.06± 0.31 | 71.33± 0.21 | 79.64± 0.23 |
| NIO | **73.17**± 0.15 | **72.79**± 0.22 | **79.76**± 0.26 |

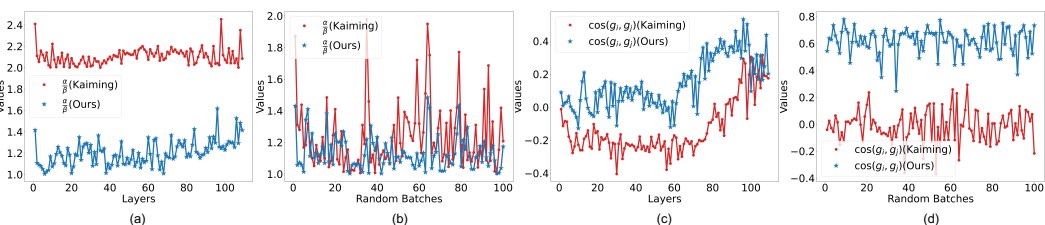

Figure 3: Empirical comparisons between the Kaiming initialization (red) and our NIO (blue) using ResNet-110 on CIFAR-10. (a) and (b) show the ratio of the maximal sample-wise gradient norm to the minimal one. (c) and (d) depict the averaged pair-wise cosine similarities between sample-wise gradients. (a) and (c) are calculated for each layer with a random batch. (b) and (d) are calculated for all parameters using 100 randomly sampled batches. Zoom in for a better view.

training setting. Each model is trained four times with different seeds. We report the average and standard deviation numbers of the accuracies on the test set.

As shown in Table 1 and 2, compared with the Kaiming initialization, we observe that MetaInit and GradInit do not achieve improvement for some cases. As a comparison, our proposed NIO consistently improves upon the Kaiming initialization on both CIFAR-10 and CIFAR-100 with and without BN. Especially for ResNet-110-BN and DenseNet-100-noBN, the performance improvements upon the Kaiming initialization are more than 0.5% on CIFAR-10 and 1.0% on CIFAR-100. Compared with GradInit, we also achieve better results by more than 1.0% for ResNet-110-noBN and DenseNet-110-noBN on CIFAR-100.

We also make some empirical comparisons between the Kaiming initialization and our NIO with ResNet-110 on CIFAR-10. As shown in Figure 3, compared with the Kaiming initialization, NIO enjoys significantly smaller ratios of the maximal gradient norm to the minimal one, and larger cosine similarities of sample-wise gradients. Therefore, after NIO, the gradient norm ratio is reduced towards 1.0, while the cosine similarity is improved towards 1.0, which indicates that the quantity in Eq. (2) is minimized, and the sample-wise gradients are close with respect to length and direction.

**Ablation Studies on Batch Setting.** We further test the sensitivity of our method to the batch setting hyper-parameters in Eq. (7), the number of sub-batches $D$ and overlap ratio $r$. We show the model performances and the time and space complexities of NIO with ResNet-110 on CIFAR-100 using different $D$ and $r$ in Tabel 3. The performances do not vary significantly with the change of $D$ and $r$. The worst accuracy (when $D = 2$ and $r = 0$ ) 75.56% in Table 3 is still higher than the Kaiming initialization result 74.54% by 1%. As $D$ increases, the time and memory consumption of NIO also grows, because more batch gradients need to be calculated by back propagation. When $D$ is small, a larger $r$ should be adopted to relieve the violation of the sample-wise hypothesis.

**Experiment Results on ImageNet.** To verify that our method can be extensible to large-scale task, we further evaluate our approach on the ImageNet dataset. Using different initialization methods,

Table 3: Model performance and complexity of ResNet-110 on CIFAR-100 with different batch settings. NIO is performed with a batchsize of 128. A smaller number of sub-batches $D$ leads to faster speed and less memory consumption, and almost does not harm the final performance. "Time" refers to the time consumed for each iteration in NIO.

| $D$ | Accuracy (%) | | | | Time | GPU Mem. |
|---|---|---|---|---|---|---|
| | $r$=0 | $r$=0.2 | $r$=0.4 | $r$=0.6 | | |
| 2 | 75.56 | 75.71 | **75.96** | 75.72 | **3.75s** | **5007M** |
| 3 | 75.60 | **76.00** | 75.97 | 75.76 | 4.07s | 6001M |
| 4 | 75.58 | **76.03** | 75.71 | 75.88 | 4.68s | 7067M |

Table 4: Top-1 accuracy of ResNet-50 on ImageNet. The results of FixUp and MetaInit are taken from their papers [58, 8], and the other methods are re-implemented by us under the same training setting. FixUp removes the BN layers as their default and the other models are trained with BN layers.

| Method | Accuracy (%) | Init. / Train time [1] |
|---|---|---|
| Kaiming [18] | 76.43±0.05 | - / 8.5h |
| FixUp [58] | 76.0 | - / 8.5h |
| MetaInit [8] | 76.0 | 0.81h / 8.5h |
| GradInit [60] | 76.50±0.05 | 0.21h / 8.5h |
| NIO (ours) | **76.71**±0.07 | 0.03h / 8.5h |

Table 5: Top-1 accuracy of Swin-Transformer (tiny) on ImageNet with and without warmup. "fail" represents that the model cannot converge. Warmup takes up the first 20 epochs when enabled.

| | Kaiming | TruncNormal | GradInit[2] | NIO (ours) |
|---|---|---|---|---|
| w/ warmup | 79.4 | 81.3 | 80.4 | **81.3** |
| w/o warmup | fail | fail | 79.9 | **80.9** |

we train ResNet-50 for 100 epochs with a batchsize of 256. Alll models are trained under the same training setting. As shown in Table 4, we improve upon the Kaiming initialization by 0.3%. We also re-implement GradInit using the default configurations in their paper [60], and have a performance close to the Kaiming initialization baseline. Moreover, NIO takes the shortest initialization time compared with the other two learning-based initialization methods, MetaInit and GradInit. It is because only 100 iterations are needed for NIO in our implementation, while MetaInit and GradInit require 1000 and 2000 iterations, respectively, as their default settings [8, 60]. Compared with the training time, the initialization time of NIO is negligible.

## 6.2 Initialization for Transformer

Vision Transformer architectures have been popular and proven to be effective for many vision tasks [11, 46, 32, 53]. But Transformer architectures suffer from unstable training due to the large gradient variance [31, 61] in the Adam optimizer [26], so have to resort to the warmup trick, which adopts a small learning rate at initialization to stabilize the training process. In order to test the effectiveness of NIO to improve the training stability, we perform NIO for Swin-Transformer [32] with and without warmup on ImageNet. Detailed training and initialization settings are described in Appendix B. As shown in Table 5, when warmup is not enabled for training, both Kaiming and Truncated normal initialized models cannot converge. As a comparison, the model with NIO achieves a top-1 accuracy of 80.9%, which is very close to the standard training result with warmup. These results indicate that NIO is able to produce a better initialization agnostic of architecture and dataset.

## 6.3 Discussions

**Why not Gradient Variance?** As indicated by Eq. (2) and Figure 3, the aim of NIO is to reduce the variability of sample-wise gradients. But why do we not directly minimize the gradient variance or just the pair-wise Euclidean distances? We point out that gradient variance is highly sensitive to gradient norm. Directly minimizing gradient variance would lead to a trivial gradient norm near zero, which disables the training. So the ability of reducing gradient variance while keeping gradient norm at a proper range is one crucial property of our proposed NIO.

---

[1]"Train time" is tested on an NVIDIA A100 server with a batchsize of 256 among 8 GPUs. "Init. time" of MetaInit and GradInit is tested for 1000 and 2000 iterations, respectively, according to the default configurations in their papers [8, 60]. NIO is performed for 100 iterations so enjoys a fast initialization process.

[2]GradInit did not perform on SwinTransformer in their paper. We adopt their initialization settings for ResNet-50 to perform GradInit on SwinTransformer and then train the model.

**Limitations and Societal Impacts.** Similar to [60], the choice of $\gamma$ has not been determined from a theoretical point yet. Calculating GradCosine accurately requires much memory consumption for large models, so it needs the batch setting relaxation. Besides, performing NIO without a dataset will be also a promising breakthrough that deserves future exploration. Our study is general and not concerned with malicious practices with respect to privacy, public health, fairness, *etc*. The proposed algorithm is lightweight and does not bring much environment burden.

## 7 Conclusion

In this paper, we propose a differentiable quantity with theoretical insight to evaluate the initial state of a neural network. Based on our sample-wise landscape analysis, we show that maximizing the quantity with the maximal gradient norm upper bounded is able to improve both training dynamic and model performance. Accordingly, we develop an initialization optimization algorithm to rectify the initial parameters. Experimental results show that our method is able to automatically produce a better initialization for variant architectures to improve the performance on multiple datasets with negligible cost. It also helps a Transformer architecture train stably even without warmup.

## Acknowledgments and Disclosure of Funding

Z. Lin was supported by the NSF China (No.s 62276004 and 61731018), NSFC Tianyuan Fund for Mathematics (No. 12026606), and Project 2020BD006 supported by PKU-Baidu Fund.

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
