# OpenReview forum: "Towards Theoretically Inspired Neural Initialization Optimization"
_NeurIPS.cc/2022/Conference — NeurIPS 2022 Accept_

### Official Review · Reviewer_u98c · 2022-07-09

**Rating:** 5
**Confidence:** 3
**Soundness:** 2 fair
**Presentation:** 3 good
**Contribution:** 2 fair

**Summary:**

This paper works on the initialization of neural network parameter based on theoretically inspired optimization algorithm. It is to substitute the previous network initialization algorithm, such as Kaiming's method, [52], etc. The proposed approach is closely related to [52] by introducing the cosine similarity of sample-wise gradient for optimizing the initial network parameter. The proposed approach is applied to Resnet, DenseNet, WideResNet, and transformer. The results show that the proposed initialization method achieved better final network training results than the Kaiming's method, and the other compared learning-based methods.

**Questions:**

Please see my questions on the novelty, experimental justifications, etc., in the weakness.

**Limitations:**

The paper clearly state the limitations of this approach.

**Strengths And Weaknesses:**

Strength:

1. The paper theoretically dig into the network initialization based on the difference of network parameter initialization and sample-wise local minima, and derive a quantity to approximate the upper-bound of training and test error bounds. The paper further approximate and minimize  the bound with constraint on gradient magnitude.  The proposed method is reasonble and inspired by theoretical analysis.

2. The proposed method is evaluated on network training and show improved results with the proposed network parameter initialization method.

Weakness:

My major questions and concerns are on the novelty over [52] and insufficient evaluation details.

1. The  theoretical analysis of this work is similar to the analysis of [52], though the conclusion of this paper is different measuring by the cosine similarity of sample-wise gradients.

2. The proposed approach is subject to an optimization problem of (6) adaptive to mini-batch based implementation. For feasibility in implementation, it makes some assumption and introduced the hyper-parameters, e.g., \lambda. How is the performence affected by the hyper-parameters of the optimization algorithm, including, e.g., \lambda, and number of iterations, etc ?

3.   Due to the randomness, e.g., the mini-batch, the performance should be reported by running a same task in several times, and the mean the variance of the performance should be more insightful to compare the different methods.

4. Because of the introduced additional computational cost in optimization, the approach is reported by running 100 iterations. How about the comparisons (speed and performance) if all the compared methods are aligned in number of running iterations for initialization?

---

> ### Author Response · Authors · 2022-07-31
> **Response to your review (part 1)**
>
> Dear Reviewer:
>
> Thanks for your valuable comments.
>
> + W1: the novelty over GradSign [52]
>
> In GradSign [52], the metric of sample-wise optima density (Eq. (1) in our paper) is proposed and proved to be an upper bound of training and generalization error. Because sample-wise optima are not tractable, they approximate it by counting the gradient sign, which is non-differentiable. The metric is used to rank neural architectures without training for neural architecture search purpose.
>
> However, the goal of our work is to develop a method for initialization optimization, instead of architecture search. So, our theoretical metric needs to be differentiable and be a function of initialization. The differences between [52] and our work are summarized as follows:
>
> (1)	The metric Eq. (1) proposed in [52] is agnostic of initialization (as shown in Figure 1 (c) and (d)). The approximated GradSign quantity is non-differentiable. So, it cannot be used for initialization optimization. In contrast, our metric Eq. (2) reflects the optimization path consistency, which is a differentiable function of initialization, and thus can serve for our initialization optimization purpose.
>
> (2)	Our method considers optimization path consistency, which is intimately related to a favorable training dynamic. The larger optimization path consistency induces a smaller gradient variance, whose benefits have been supported by prior studies [28,54]. See more details in Lines 174-184 in our paper. As a comparison, [52] does not enjoy this advantage.
>
> (3) The analyses of our work and [52] are similar because both metrics can be the upper bound of training and generalization error. They both rely on similar assumptions and Lemma1. But the proof details are different. We need to consider the sample-wise optimization path ($\theta^*_i-\theta_0$), while [52] is only concerned with sample-wise optima $\theta^*_i$.
>
> (4)	Although both metrics are related to model performance, [52] and we use them for different purposes. The other parts of our work, including our GradCosine quantity, the objective of our neural initialization optimization, and the algorithm to solve it, are original and have no overlap with [52].
>
> So, we think it is not fair and reasonable to deny our novelty and contribution just because both our work and [52] have a metric that can be the training and generalization error bound. They are studies for different purposes.
>
> + W2: How is the performance affected by the hyper-parameters, including $\\lambda$ and the number of iterations?
>
> Our algorithm introduces three hyperparameters, the number of sub-batches $D$, the overlap ratio $r$, and the upper bound constraint of gradient norm $\\gamma$. We do not have a hyperparameter denoted as $\\lambda$ in our paper. We think what you refer to is $\\gamma$. We have made ablation studies for the hyperparameters $D$ and $r$, see Table 3 in our paper. For $\\gamma$, we just follow the choices suggested by GradInit [53] because it plays a similar role in our work and GradInit [53]. It ensures that the gradient norm at initialization for training will not be too large or too small. We list the performances of NIO using ResNet-50 on ImageNet with different $\\gamma$ as follows:
>
> | $\\gamma$ | 1 | 5 | 10 | 15 | 20 | 30 |
> |---|---|---|---|---|---| --- |
> |NIO| 76.25 | 76.57 | 76.71 | 76.76 | 76.63 |76.68 |
>
> We run NIO for 100 iterations on ImageNet because we find that more iterations do not bring significant improvement. The effect of iteration is shown as follows:
>
> | iteration | 50 | 75 | 100 | 150 | 200 | 500 |
> |---|---|---|---|---|---|---|
> |NIO | 76.44 | 76.67 | 76.71 | 76.70 | 76.74 | 76.65 |
>
> It is shown that as long as the hyperparameter $\\gamma$ or the number of iterations is within a proper range, the final performance will not deviate too much.
>
> ----
> Reference
>
> [28] L. Liu, H. Jiang, P. He, W. Chen, X. Liu, J. Gao, and J. Han. On the variance of the adaptive learning rate and beyond. In ICLR, 2020.
>
> [52] Z. Zhang and Z. Jia. Gradsign: Model performance inference with theoretical insights. In ICLR, 2022.
>
> [53] C. Zhu, R. Ni, Z. Xu, K. Kong, W. R. Huang, and T. Goldstein. Gradinit: Learning to initialize neural networks for stable and efficient training. In NeurIPS, 2021.
>
> [54] J. Zhuang, T. Tang, Y. Ding, S. C. Tatikonda, N. Dvornek, X. Papademetris, and J. Duncan. Adabelief optimizer: Adapting stepsizes by the belief in observed gradients. In NeurIPS, 2020.

---

> > ### Author Response · Authors · 2022-07-31
> > **Response to your review (part 2)**
> >
> > + W3: Mean and variance of the performance should be more insightful to compare the different methods.
> >
> > We indeed run a same task for multiple times with different seeds. Each model is trained for four times as described in Line 250 in our paper. We report the means and standard deviations of results on CIFAR-10 and CIFAR-100, see Tables 1 and 2 in our paper. For experiments on ImageNet, the randomness is not as notable as that on CIFAR. Multiple experiments have close performances. So, we did not report the standard deviation in Tables 4 and 5. In order to relieve your concern, we report the means and standard deviations of GradInit and NIO results in Table 4 as follows:
> >
> > | Method | Accuracy (%)|
> > | --- | --- |
> > | GradInit |   76.50$\\pm$ 0.05|
> > | NIO |          76.71$\\pm$ 0.07|
> >
> > The means and standard deviations of SwinTransformer results in Table 5 are shown as follows:
> >
> > |  | Kaiming | TruncNormal | NIO (ours) |
> > | --- | --- | --- | --- |
> > | w/ warmup | 79.44$\pm$0.03 | 81.28$\pm$0.03 | 81.30$\pm$0.05|
> > | w/o warmup| fail | fail | 80.91$\pm$0.06|
> >
> >
> >
> > + W4: How about the comparisons if all methods are aligned in number of running iterations for initialization?
> >
> >
> > In Table 4, our method is performed for 100 iterations, while GradInit requires 2000 iterations as suggested by their paper. We reimplement our method for 2000 iterations, and GradInit for 100 iterations. The results of performance and speed are shown as follows:
> >
> > | Init. Time | 100-iter | 2000-iter |
> > |---|---|---|
> > | GradInit | 0.01 | 0.21 |
> > | NIO | 0.03 | 0.6 |
> >
> > |Top-1 Acc. | 100-iter | 2000-iter |
> > |---|---|---|
> > |GradInit | 75.96 | 76.50 |
> > |NIO | 76.71 | 76.68 |
> >
> > It is shown that our method has close performances for 100 and 2000 iterations. In contrast, GradInit has a decreased performance when performing for only 100 iterations. Although the speed of our method per iteration is slower than GradInit, we have a faster convergence and only require a less number of iterations for initialization.

---

> > > ### Comment · Reviewer_u98c · 2022-08-08
> > > **Further comments**
> > >
> > > Thanks for the responses to my concerns. The authors mostly clarified my concerns on novelty, effect of hyper-parameters, number of runs to report results, and results with aligned number of iterations. I am fine to increase score and tend to accept, and suggest that the revised paper should include these discussions and new results.

---

> > > > ### Author Response · Authors · 2022-08-09
> > > > **Thanks for appreciating our work**
> > > >
> > > > We are glad to know that our response has addressed your concerns. We would like to thank you for appreciating our work and the constructive suggestions. We will include the discussions and results in our revised paper according to your suggestions.

---

### Official Review · Reviewer_ugxF · 2022-07-10

**Rating:** 6
**Confidence:** 4
**Soundness:** 2 fair
**Presentation:** 3 good
**Contribution:** 3 good

**Summary:**

This work proposes a new initialization method for neural network. The authors first use Fig. 1 to illustrate drawbacks of the sample-wise local optima density $\Phi_{S,l}$ that adopts Manhattan distance between the pair-wise local optima, and lead to a Cosine similarity of sample-wise local optima $\Theta_{S,l}$. Then the authors introduce to approximate local optimum by one step gradient descent and approximate $\Theta_{S,l}$ by Eq. 4 (GradCosine). Finally, the initialized weights $\theta_0$ are obtained by maximizing GradCosine and GradNorm. Extensive experiments are conducted to verify the efficacy of this approach, showing that GradCosine surpasses MetaInit and GradInit on multiple datasets in both accuracy and speed.

**Questions:**

1. The authors claim that (Line 114-115): The optimization path from the initialization to the local optima of the two samples in Fig. 1(d) are more consistent compared to that in Fig. 1(c). However, I notice that $\theta_0$ in Fig. 1(c) is closer to $\theta_1^*, \theta_2^*$ than $\theta_0$ in Fig.1 (d), resulting in a larger angle $\angle(\theta_1^*-\theta_0, \theta_2^*-\theta_0)$. My point is that the angle cannot reflect the quality of optimization path since an initial point that is closer to the optimal point might as a larger angle.
2. I have one concern on the approximation in Eq. 3. For initialized weights $\theta_0$, such an approximation seems not reasonable. I hope the authors could discuss the rationality of it.


**Limitations:**

The limitations are addressed.

**Strengths And Weaknesses:**

Strengths:
1) The writing is logical and fluent.
2) The experimental results are good.
3) The motivation is interesting. Nevertheless, I have one question about it, please see the first point in Question.

Weakness (Question):
1) In Line 28, the authors claim that “these methods merely use the first order training dynamic as the main optimization objective”. However, MetaInit[8] and GradInit[53] both adopts gradient descent method.
2) It seems that the theoretical analysis in Sec. 3.2 cannot show that $\Theta_{S,l}$ has a tighter bound than $\Phi_{S,l}$. Then I think Sec. 3.2 is a little redundant, which can be put into the supplementary.
3) Typo: ‘GradCoisne’ in Line 6.

---

> ### Author Response · Authors · 2022-07-31
> **Response to your review**
>
> Dear Reviewer:
>
> Thanks for your valuable comments.
>
> + W1: Both MetaInit and GradInit adopt gradient descent method.
>
> Yes, the objective of MetaInit is also a function of the model gradient. The objective of GradInit requires to perform a step of gradient descent. Their adopted gradient is the full gradient of the whole CE loss. As a comparison, we focus on the similarity of sample-wise gradients, which decompose the whole CE loss into each sample. Our objective has theoretical supports to be related to model performance. We will rephrase the sentence in Line 28 to make the comparison clear and avoid confusion.
>
> + W2: Sec 3.2 is a little redundant and can be put into the supplementary.
>
> $\\Psi$ in Eq. (1) is agnostic of initialization. It is approximated by a non-differentiable quantity (counting gradient signs) in GradSign [1] to rank and search neural architectures. In contrast, our proposed $\\Theta$ in Eq. (2) reflects the optimization path consistency, which is a differentiable function of initialization and thus can serve for our initialization optimization purpose. Actually, we do not compare the tightness between $\\Theta$ and $\\Psi$. We want to show that both $\\Theta$ and $\\Psi$ can be the upper bound of training and generalization error and are related to model performance. But our metric $\\Theta$ is suitable for initialization optimization, while $\\Psi$ cannot serve for this purpose.
>
> We will follow your suggestion and adjust the organization to avoid confusion.
>
> + W3: Typo in lilne 6.
>
> Thanks for rectifying us. We will fix the typo.
>
> + Q1: The initial point with a larger angle is closer to the global optimum.
>
> In Figure 1 (c) and (d), it seems that an initialization with a smaller cosine similarity (larger angle) is closer to the global optimum of the two samples. But we should note that:
>
> (1)	The optimum of a given network is not unique. It is dependent on its initialization. The model with different initialization points will converge to different optimal parameters, even though their performances may be close. So, what we do here is looking for an initialization whose sample-wise optimization paths are more consistent (smaller angle), instead of the inverse way--looking for a point closer to the global optimum of some given sample-wise optima and landscapes, which is impossible for a real network. The larger optimization path consistency induces a smaller gradient variance, whose benefits have been supported by prior studies [2,3]. See more details in Lines 174-184 in our paper.
>
> (2)	The illustration in Figure 1 is a toy example with only two samples and simple landscapes. In a real network, the landscapes are highly non-convex in a high dimension, and there are a larger number of training samples. In this case, the global optimum does not necessarily lie in the convex hull. The point with a larger angle to sample-wise optima is not ensured to be closer to the global optimum.
>
> In conclusion, an initialization with more consistent sample-wise optimization paths is more favorable. What we do is looking for such initialization, instead of the global optimum of given sample-wise landscapes. Figure1 (c) and (d) are toy examples that are only used to show that the metric Eq. (1) is agnostic of initialization, while ours Eq. (2) reflects the optimization path consistency and is a function of initialization. We will make more explanation for Figure 1 in our revised version to avoid confusion.
>
>
> + Q2: Discuss the rationality of the approximation in Eq. (3).
>
> Note that in Eq. (3) $\\theta^*_i$ is not the global optimum. It is the local optimum for sample $i$. If we optimize a model on only one training sample, it is very easy to finish the training. Only several iterations are needed to attain a zero loss. So, we make the first-order approximation for the sample-wise optimization, i.e., the sample-wise optimum can be reached via only one step of gradient descent.
>
>
> ----
> Reference
>
> [1] Z. Zhang and Z. Jia. Gradsign: Model performance inference with theoretical insights. In ICLR, 2022.
>
> [2] L. Liu, H. Jiang, P. He, W. Chen, X. Liu, J. Gao, and J. Han. On the variance of the adaptive learning rate and beyond. In ICLR, 2020.
>
> [3] J. Zhuang, T. Tang, Y. Ding, S. C. Tatikonda, N. Dvornek, X. Papademetris, and J. Duncan. Adabelief optimizer: Adapting stepsizes by the belief in observed gradients. In NeurIPS, 2020.

---

> > ### Comment · Reviewer_ugxF · 2022-08-08
> > **Thanks for the response**
> >
> > Thanks for your response.
> >
> > I am still confused about the claim that smaller angle between initialization and optima indicates more consistent sample-wise optimization paths. I wonder what is the definition of the consistency of sample-wise optimization path and could you express the relationship between it and the angle in mathematics? Or could you intuitively explain the relationship?
> >
> > Overall, the rebuttal resolves most of my concerns, except the above one. I tend to maintain my original score and would like to raise the score if the authors could address my confusion.

---

> > > ### Author Response · Authors · 2022-08-08
> > > **Explaination on the consistency of sample-wise optimization**
> > >
> > > Thanks for your reply.
> > >
> > > Sample-wise optimization refers to training the model only on each single sample. So, the optimization path for sample $i$ can be characterized by $\\theta^*_i-\\theta_0$, where $\\theta_0$ is the initialized point, and $\\theta^*_i$ is the converged optimum of only training on sample $i$. Because all sample-wise optimization paths have the same starting point $\\theta_0$, we use the averaged Cosine similarity to measure their consistency. The consistency of sample-wise optimization path can be formulated as:
> > > $\\frac{1}{n^2}\\sum_\{ij\}\\cos\\angle(\\theta^*_i-\\theta_0, \\theta^*_j-\\theta_0)$. So, if the angle between the paths from initialization $\\theta_0$ to each local optimum $\\theta^*_i$ is small, we will have more consistent sample-wise optimizaiton paths. It is approximated by our GradCosine in Line 6 in Algorithm1.
> > >
> > > It is shown that our proposed Eq. (2) reflects the optimization path consistency and is a function of initialization. Our aim is to look for a $\theta_0$ that minimizes Eq. (2).
> > >
> > >
> > > Please let us know if anything still confuses you. We will make these definitions clearer in the revised paper.

---

> > > > ### Comment · Reviewer_ugxF · 2022-08-09
> > > > **Thanks for the response**
> > > >
> > > > Thanks for your response.
> > > >
> > > > The explanation addresses most of my concerns. I would like to raise the score to 6. I hope the authors could clarify the definition of optimization path and the definition of consistency in the next version.
> > > >
> > > > Nevertheless, I am still a little worried about the first-order approximation in Eq.3, though it is a commonly-used technique in bi-level optimization methods. While, such an approximation can be biased for the initialized weights. I hope the authors could further discuss it in the next version (or list it in the limitation).

---

> > > > > ### Author Response · Authors · 2022-08-09
> > > > > **Thanks for appreciating our work**
> > > > >
> > > > > We are glad to know that our response has addressed most of your concerns. We would like to thank you for appreciating our work and the constructive suggestions.
> > > > >
> > > > > Because sample-wise optima are intractable, we rely on the approximation in Eq. (3) to construct a tractable optimization objective. If we use Eq. (3) to optimize for $\\theta^*_i$, the bias may cause an inaccurate solution. But our aim is to look for an initialization. Eq. (3) approximates $\\theta_i^*-\\theta_0$ as $-\\eta g_i$, so turns the objective from optimization path consistency into gradient consistency. It does not deviate from our goal for an initialization. Besides, prior studies also indicate that low gradient variance at initialization is favorable.
> > > > >
> > > > > We will discuss it and revise our paper according to your suggestions.

---

### Official Review · Reviewer_JX6Q · 2022-07-11

**Rating:** 6
**Confidence:** 4
**Soundness:** 3 good
**Presentation:** 3 good
**Contribution:** 3 good

**Summary:**

The paper first introduces a quantity, the cosine similarity of sample-wise local optima to evaluate the model performance at the initialization. They theoretically proved that their proposed quantity is the upper bound of both the training and generalization error under certain assumptions. Based on this theoretical finding, they approximate the sample-wise optimum with the first-order approximation to make the quantity differentiable and tractable. As a result, they simplify the upper bound quantity and achieve the initialization by maximizing the quantity with the gradient-based method. Their empirical results show that they can achieve better performance on various datasets and network structures compared to other initialization methods.

**Questions:**

1. Can we have some experiments to show that the proposed cosine similarity of sample-wise local optima is more suitable for evaluating the initialization quality?

2. Does the initialization framework work for multiple optimization methods? For example, like sharpness-aware-minimization?

3. It seems the method is dataset-dependent, will we have the same benefits if we initialize the parameter of the networks with a different dataset? For example, training on cifar-10 but initialized with cifar100 or ImageNet?

4. Does this only work for initialization? Can we repeatedly use it during the training and will that be better?


**Limitations:**

The authors fairly addressed the limitations and potential negative societal impact of their work.

**Strengths And Weaknesses:**

Strengths:
The paper is well organized and easy to follow. The theoretical analysis seems correct and motivated. Their empirical results are quite good, especially on CIFAR datasets.

Weaknesses:
It will be better if we can have some experiments to show that the proposed cosine similarity of sample-wise local optima is useful or better than the previous methods. Since the initialization method is motivated by this quantity, it’s better to make the usefulness of this quantity clear.

The citation in Appendix A for the proof of Lemma 1 seems wrong.

---

> ### Author Response · Authors · 2022-07-31
> **Response to your review (part 1)**
>
> Dear Reviewer:
>
> Thanks for your valuable comments.
>
> + W1 and Q1: Can we have experiments to show the proposed metric is better?
>
> The density of sample-wise local optima $\\Psi$ in Eq. (1) is agnostic of initialization. It is approximated by a non-differentiable quantity (counting gradient signs) in GradSign [1] to rank neural architectures without training for neural architecture search purpose. As a comparison, our proposed cosine similarity of optimization paths $\Theta$ in Eq. (2) reflects the optimization path consistency (cosine similarity between $\\theta_i^*-\\theta_0$ and $\\theta_j^*-\\theta_0$). It is a differentiable function of initialization $\\theta_0$, and thus can serve for our initialization optimization purpose. Although $\\Psi$ in Eq. (1) is also related to model performance as adopted in [1], it is used to evaluate different architectures agnostic of initialization and cannot reflect initialization quality. Our $\\Theta$ in Eq. (2) can evaluate a model under different initializations.
>
> In order to quantitatively compare the metric $\\Psi$ in Eq. (1) with our proposed $\\Theta$ in Eq. (2), we train ResNet-110 on CIFAR-100 with 10 different initializations $\\theta_0^{(1)},\\cdots,\\theta_0^{(10)}$, so we have 10 trained models $M^{(1)},\\cdots,M^{(10)}$, and their accuracies $Acc^{(1)}, \\cdots, Acc^{(10)}$. For each model $M^{(i)}$, we select out the wrongly classified training samples, and finetune the model on each of these samples until the sample is correctly predicted. So, we have the sample-wise optimal models $\\{M^{(i)}_j\\}$, $j=1,…,\\mathcal{J}_i$, where $\\mathcal{J}_i$ is the number of wrongly classified training samples of model $M^{(i)}$. And then we calculate the quantities $\\Psi$ and $\\Theta$ using these sample-wise optimal model parameters according to Eq. (1) and Eq. (2) ($\\mathcal{H}$ is set as 1), respectively. We get the estimated quantities $\\Psi^{(1)},\\cdots,\\Psi^{(10)}$ and $\\Theta^{(1)},\\cdots,\\Theta^{(10)}$. Finally, we calculate Kendall score $\\tau$ between $\\{Acc^{(i)}\\}$ and $\\{\\Psi^{(i)}\\}$, and between $\\{Acc^{(i)}\\}$ and $\\{\\Theta^{(i)}\\}$. The score ranges from -1 to 1 and is able to evaluate rank correlation of data pairs. $\\tau=1$ when the rankings are identical, and $\\tau=-1$ when the rankings are reversed. If the rankings have a low correlation, $\\tau$ is near 0. We have their values as follows:
>
> | Metric | Kendall $\\tau$ |
> | --- | --- |
> | $\\Psi$ (Eq. (1)) |   -0.28|
> | $\\Theta$ (Eq. (2)) |   -0.73|
>
> It reveals that $\\{Acc^{(i)}\\}$ are significantly inversely correlated with $\\{\\Theta^{(i)}\\}$, while $\\{Acc^{(i)}\\}$ and $\\{\\Psi^{(i)}\\}$ show a low correlation. So, $\\Theta$ is more preferable to evaluate a model under different initializations.
>
> + W2: The citation in Appendix A for the proof of Lemma 1 is wrong.
>
> Yes, thanks for rectifying us. We will correct the wrong citation.
>
> + Q2: Does the initialization framework work for other optimization methods, like sharpness-aware-minimization?
>
> Both SGD and Adam can be adopted to solve our initialization optimization problem (Eq. (8)). The advanced technique sharpness-aware-minimization is integrated on a loss function to induce better generalization. However, our objective in Eq. (8) is specified by the gradient cosine and gradient norm quantities. It is not a loss function at all that measures the error of a model. So, it has no access to the landscape of a loss function, let alone the sharpness. Our goal is to look for a better initialization, while generalization is more affected by the loss function and the optimizer for training.
>
> + Q3: Can we have the same benefits if we initialize the networks with a different dataset?
>
> Thanks for the interesting question. Our method is indeed aware of architecture and dataset. We perform our NIO with CIFAR-100 on a ResNet-110 and train the initialized model (with a new fc classification layer due to different numbers of classes) on CIFAR-10. We observe no significant benefit. The final performance is around the baseline result without NIO. We guess it is because that the gradient patterns on different datasets are different. Our objective of initialization is supervised. Its optimization on CIFAR-100 would be invalid when training on CIFAR-10 with a different label space. The exact dependency of our method on the dataset needs more exploration. Future work such as unsupervised learning-based initialization may relieve the dependence on dataset.
>
> Despite the dependence, we think it does not impede the practical implementation. We only need a small portion of the dataset. The cost of our initialization method is also friendly. So, it is not necessary to initialize with a small dataset and train on a large dataset.
>
> ----
> Reference
>
> [1] Z. Zhang and Z. Jia. Gradsign: Model performance inference with theoretical insights. In ICLR, 2022

---

> > ### Author Response · Authors · 2022-07-31
> > **Response to your review (part 2)**
> >
> > + Q4: Can we repeatedly use it during training for better performance?
> >
> > Using our method during training does not contribute to the final performance. When we start training, the training and generalization abilities are mainly decided by the training loss and optimizer. Our objective is used to look for a better initialization. It just optimizes the starting point. During training, the parameter update direction should come from the signal of a loss function. But our objective of initialization is not a loss function. Besides, out method only tunes the variance scaling ($M$ in our paper line 196) of the parameters. Changing the scaling of parameters in each layer during training will surely deviate the original optimization.

---

### Official Review · Reviewer_Jy3Y · 2022-07-11

**Rating:** 6
**Confidence:** 4
**Soundness:** 3 good
**Presentation:** 3 good
**Contribution:** 3 good

**Summary:**

Authors study the problem of finding (learning) the best neural network initialization, and propose GradCosine - a measure of fitness for neural network initialization, based the similarity between individual sample gradients. This measure can then be optimized using and iterative procedure denoted as NIO (neural initialization optimization). Authors explain how GradCosine relates to model training and show a the relation between this quantity and the density of sample-wise local optima ([52] in the paper).
From a theoretical point of view, the paper demonstrates that minimizing GradCosine results in favourable bounds on the training loss. For practical applications, authors offer a sub-batch version of GradCosine that can be computed more efficiently.

**Questions:**

Note: the question / suggestion below bears little significance. If authors are constrained by response time, please ignore it.

To the best of my understanding, even batched GradCosine currently requires more GPU memory than the objectives of MetaInit or GradInit (per model parameter). This could be a limitation when applying NIO for large models, e.g. transformer language models.
I wonder if it is possible to reduce the memory usage algorithmically.

Suppose that most of that memory is contributed by gradients w.r.t. weight matrices in linear layers, whether in convolutions or transformer projections.

For linear layers, one can observe that __one-sample gradients w.r.t. weight matrices are low-rank__.
In the simplest case, an MLP linear layer with batch size 1 will always receive rank-1 gradient - due to the fact that the gradient w.r.t. weight matrix is a product of two matrices of shape (1 x in_features) and (out_features x 1), where both "1"s correspond to batch size.
For convolution and attention layers, this will similarly result in low-rank gradients.

This trick is described in more detail in [1], though i believe that it was invented prior to that work.

Put simply, if your gradient w.r.t. weight matrix was computed with batch size 1, you can compute the pairwise products for GradCosine without computing the gradients w.r.t. weight, using only the gradients w.r.t. activations. Furthermore, you could avoid storing activations for all layers by re-computing them on the fly[2].


[1] https://arxiv.org/abs/2110.11309

[2] https://arxiv.org/abs/1604.06174v2




**Limitations:**

To the best of my knowledge, authors have sufficiently addressed the limitations of their work.
As for the societal impact, this specific paper contains fundamental research in deep learning, thus it is hard to foresee its societal impact.

**Strengths And Weaknesses:**

To the best of my understanding, the theoretical analysis appears sound, based on reasonable assumptions. [That said, I am no expert in the area of learned initialization.] The main experiments look reasonable, though I would recommend making some extra comparisons.

1. in Table 5, you evaluate how NIO trains SWIN transformer without warmup. At least one baseline (GradInit[53]) also claims to work in that setting. Perhaps it would be best to compare NIO against that.

2. Table 4 does not report standard deviations, while all prior experiments do. It might be useful to include them - or explain why they are missing.


On an unrelated note, I must applaud authors for supplying a Dockerfile in their supplementary code. Publishing the corresponding docker container will make it easier for future researchers to reproduce this work and build on it, even if the required libraries break compatibilities.


### Typos / nitpicking

> L215  making the sub-batches overlapped with each other **stables** the optimization

perhaps a typo? "stables" -> "stabilizes"


> L250 Each model is trained **for** four times with different seeds.

consider removing "for"


> NIO is able to produce a better initialization that benefits model performance agnostic of architecture and dataset.

[nit] this is concluded at an early section, where the only evidence is training cnn-only models on CIFAR-10/100. As a weak suggestion, I would recommend making this conclusion later, once you demonstrate NIO performance for transformers and ImageNet.

---

> ### Author Response · Authors · 2022-07-31
> **Response to your review**
>
> Dear Reviewer,
>
> Thanks for your valuable comments.
>
> + W1: Compare NIO with baseline (GradInit) on Swin Transformer without warmup.
>
> GradInit tests on a language Transformer instead of vison Transformer in their paper, so we did not compare with GradInit in Table 5. We adopt their initialization settings for ResNet-50 on ImageNet to perform GradInit on SwinTransformer and then train the model with and without warmup. The results are shown as follows:
>
> |  | Kaiming | TruncNormal | GradInit | NIO (ours) |
> | --- | --- | --- | --- | --- |
> | w/ warmup | 79.4 | 81.3 | 80.4 | 81.3 |
> | w/o warmup| fail | fail | 79.9 | 80.9 |
>
> We observe that GradInit also successfully trains SwinTransformer without warmup, but has a lower performance than ours. We will add this result in our revised version.
>
> + W2: Table 4 does not report standard deviations like prior results.
>
> Usually experiments on small datasets such as CIFAR are sensitive to randomness and have results with fluctuation. Repeated experiments on ImageNet classification usually have close performances with a small deviation. So, we did not report the standard deviations in Table 4. In order to relieve your concern, we report the means and standard deviations of GradInit and NIO results in Table 4 as follows:
>
> | Method | Accuracy (%)|
> | --- | --- |
> | GradInit |   76.50$\\pm$ 0.05|
> |NIO |          76.71$\\pm$ 0.07|
>
> + W3: Typos and writing suggestions
>
> Thanks for rectifying us. We will correct the typos and modify our paper according to your suggestions.
>
> + Q1: Memory consumption of computing GradCosine
>
> When the input image/sentence size is large, the memory consumption of computing GradCosine is a problem. Our simple solution is to adopt a small batchsize when performing NIO. The GradCosine quantity can be optimized easily. It usually gets saturated (>0.9) after tens of iterations. So, we can use a small batchsize and train for more iterations. It nearly does not harm the initialization quality and the final performance.
>
> Thanks for your guidance and reminding us of the references [1,2] that could potentially improve our method. Using the low rankness of one-sample gradients w.r.t weight matrices to reduce memory overhead is interesting. If we can efficiently get the decomposed matrices, the memory overhead will be reduced greatly, and the pair-wise inner-products of gradients can also be computed efficiently. We think there are some issues that need to be considered.
>
> We notice that [1] adopts decomposed matrices as a proxy to optimize the editor parameters, and thus reduces the computational and memory cost. However, our method requires the second-order gradient to optimize our objective in the same neural network (instead of another network as adopted in [1]). In this case, it is not sure whether optimizing the similarity between the sample-wise gradients decomposed into (1, in_channles) and (1, out_channels) is still a good approximation.
>
> Besides, it is not sure whether the checkpointing technique (pytorch implementation of [2], please refer to https://pytorch.org/docs/stable/checkpoint.html) can be easily introduced to our optimization algorithm, since the second-order gradient requires the storage of both gradients and activations and has more complicated dependencies leading to a complex topological structure that is hard to free some memories on the fly.
>
> As a future work, we will think carefully how to improve our method from both the theoretical (e.g. [1]) and systematic (e.g. [2]) perspectives.
>
> ----
> References
>
> [1] Fast Model Editing at Scale, Mitchell et al., ICLR 2022.
>
> [2] Training Deep Nets with Sublinear Memory Cost, Chen et al., arXiv:1604.06174v2

---

> > ### Comment · Reviewer_Jy3Y · 2022-08-09
> > **Quick Acknowledgement**
> >
> > Thank you for the clarifications and extra evaluation. If possible, I would also appreciate taking a look at a revised version, if you can push it before the revision deadline. However, this is a weak request and will fully understand if authors do not find time for that..

---

> > > ### Author Response · Authors · 2022-08-09
> > > **A summary of revisions that will be made**
> > >
> > > Thanks for appreciating our work. We are sorry that there are only a few hours left before the discussion deadline. We will be in a hurry for the revision version because we also need to consider how to fit in 9 pages after the revision and adding more results and discussions. But we would like to summarize the revisions that remain to be done as follows:
> > >
> > > (1) We will add the standard deviations for the results in Tables 4 and 5. We will also include the results of GradInit on Swin Transformer with and without warmup in Table 5. The sentence that "These results indicate that NIO is able to produce a better initialization that benefits model performance agnostic of architecture and dataset" (Lines 258-259) will be removed into the end of Sec. 6.2.
> > >
> > > (2)	As responded to Reviewer JX6Q, we may include the simple experiment of calculating the Kendall score to show that our Eq. (2) is more preferable to evaluate initialization quality.
> > >
> > > (3)	As responded to Reiviewer ugxF, we will make clearer definitions for “optimization path” and “optimization path consistency”. We will also include the discussions about the rationality and necessity of the approximation in Eq. (3).
> > >
> > > (4)	As suggested by Reviewer ugxF, we will rephrase the sentence when describing MetaInit and GradInit in Line 28 to avoid misunderstanding. We will also add more explanations for Figure 1 to avoid confusion.
> > >
> > > (5)	As suggested by Reviewer u98c, we think it is better to itemize the novelties over GradSign in the Related Work section. We will also include the ablation studies on the hyper-parameter $\\gamma$ and the number of iterations, and compare with GradInit when both are performed for the same iterations.
> > >
> > > (6)	All typos will be fixed.
> > >
> > > Other constructive suggestions and discussions will also be considered by us.

---

### Meta-Review · Area_Chair_r2Zc · 2022-08-26

**Recommendation:** Accept
**Confidence:** Less certain

**Metareview:**

The paper introduces a new procedure to initialize the optimisation in training process of DNN models, including the recent ViT architecture. All the reviewers recommend acceptance and appreciate the promising empirical results backed by the strong theoretical foundations. AC recommends acceptance as well.

**Award:**

No

---

### Decision · Program_Chairs · 2022-09-14

Accept